# Medial thalamic stroke and its impact on familiarity and recollection

Lola Danet[1,2,3]*, Jérémie Pariente[1,3], Pierre Eustache[1], Nicolas Raposo[1,3], Igor Sibon[4], Jean-François Albucher[1,3], Fabrice Bonneville[1,3], Patrice Péran[1], Emmanuel J Barbeau[2]

[1]Toulouse NeuroImaging Center, Université de Toulouse, Inserm, Toulouse, France; [2]Brain and Cognition Research Centre, CNRS, University of Toulouse Paul Sabatier, Toulouse, France; [3]Neurology Department, CHU Toulouse Purpan, Toulouse, France; [4]Department of Diagnostic and Therapeutic Neuroimaging, University of Bordeaux, Bordeaux University Hospital, Bordeaux, France

**Abstract** Models of recognition memory have postulated that the mammillo-thalamic tract (MTT)/anterior thalamic nucleus (AN) complex would be critical for recollection while the Mediodorsal nucleus (MD) of the thalamus would support familiarity and indirectly also be involved in recollection (Aggleton et al., 2011). 12 patients with left thalamic stroke underwent a neuropsychological assessment, three verbal recognition memory tasks assessing familiarity and recollection each using different procedures and a high-resolution structural MRI. Patients showed poor recollection on all three tasks. In contrast, familiarity was spared in each task. No patient had significant AN lesions. Critically, a subset of 5 patients had lesions of the MD without lesions of the MTT. They also showed impaired recollection but preserved familiarity. Recollection is therefore impaired following MD damage, but familiarity is not. This suggests that models of familiarity, which assign a critical role to the MD, should be reappraised.
DOI: https://doi.org/10.7554/eLife.28141.001

*For correspondence:
lola.danet@inserm.fr

Competing interests: The authors declare that no competing interests exist.

## Introduction

A number of studies have been carried out on thalamic amnesia, with the aim of clarifying the role of thalamic nuclei and bundles in memory processes (*von Cramon et al., 1985*; *Cipolotti et al., 2008*; *Carlesimo et al., 2011*; *Pergola et al., 2012*). Dense pathways link the medial temporal lobe to the anterior part of the thalamus (*Aggleton and Brown, 1999*; *Aggleton et al., 2011*). More precisely, the Mamillothalamic tract/Anterior thalamic nucleus (MTT/AN) complex is thought to be critical for memory because of its direct and indirect connections with the hippocampus (*Ghika-Schmid and Bogousslavsky, 2000*; *Van der Werf et al., 2000*; *Aggleton et al., 2011*; *Edelstyn et al., 2012*). The mediodorsal (MD) may also play a role in memory, because of its direct connections with anterior subhippocampal structures, most notably the perirhinal cortex (*Aggleton et al., 2011*). An influential dual-process model suggested that the AN/MTT complex is critical for recollection, the ability to retrieve part of the experience associated with a stimulus, while the MD is important for familiarity, a simpler process related to the mere feeling that the stimulus has been experienced before (*Aggleton and Brown, 1999*). Contrary to single-process theories that state that recollection and familiarity map on to strong and weak memories, this model therefore assumed that these two processes are functionally and anatomically independent.

However, subsequent findings did not fully support this simple anatomical-functional dissociation. Although patients with AN lesions have impaired recollection, they also usually have lesions of other diencephalic structures, which sometimes hampers interpretation of the results (reviewed in *Aggleton et al., 2011*). Thus, a recurrent problem is that the AN's role in recognition memory is

often deduced from lesions to adjacent afferent structures, such as the mammillary bodies or the MTT, in the absence of specific AN damage (*Carlesimo et al., 2007*; *Tsivilis et al., 2008*; *Vann et al., 2009*). Some results also appear to contradict the model's predictions. For example, *Cipolotti et al. (2008)* reported the case of two patients who both had damage to the left AN/MTT and MD. One of the patients also had damage to the right AN/MTT (and lateral dorsal nucleus), while the other had damage to the right MD. According to Aggleton and Brown's model, these right-sided lesions should have meant that, for visual material, familiarity should have been preserved in the first patient and recollection should have been preserved in the second. However, these predictions were not borne out. Furthermore, in some patients with an AN lesion, familiarity is also impaired, albeit to a lesser extent than recollection (*Kishiyama et al., 2005*).

Similarly, we have yet to pinpoint the role of the MD in memory (*Edelstyn et al., 2012*; *Cipolotti et al., 2008*; *Pergola et al., 2012*; *Tu et al., 2014*). Experimental, selective, lesions of the medial region of the thalamus induced recognition memory impairment in nonhuman primates. It was hypothesised that they could more precisely be related to lesions of the magnocellular part of the MD (*Aggleton and Mishkin, 1983a*, *Aggleton and Mishkin, 1983b*; *Parker et al., 1997*). However, the magnitude of the impairment was moderate compared to direct lesions of the perirhinal cortex. Indeed, *Aggleton and Brown (1999)* noted that there could be other output routes from the perirhinal cortex to the rest of the brain than only through the MD. Furthermore, recordings in the MD (and in the paraventricular midline thalamic nuclei as well) in nonhuman primates revealed neurons that were sensitive to repetition, apparently supporting the view that this nucleus could be in involved in memory processes (*Fahy et al., 1993*).

In the human, *Zoppelt et al. (2003)* assessed recollection and familiarity in a group of five patients with MD lesions (three right, two left). These patients exhibited impairment of both processes, prompting the authors to argue for a role of the MD in recollection. *Soei et al. (2008)* reported impaired relational memory in six patients with MD damage (three left, two right, one bilateral). However, none of them exhibited nonrelational memory impairment, suggesting overall impaired recollection but preserved familiarity after MD damage. Recent studies using more refined imaging approaches to localize lesions have corroborated the idea that MD damage results in a recollection deficit (*Pergola et al., 2012*; *Tu et al., 2014*). By contrast, *Edelstyn et al. (2016)* described in a case study a patient with right MD damage who had a more pronounced deficit of familiarity than of recollection. This study followed two fMRI studies by the same group, which had evidenced activation of the MD in relation to familiarity (*Montaldi et al., 2006*; *Kafkas and Montaldi, 2014*). The MD may therefore play a role in recollection despite the prediction made by *Aggleton and Brown (1999)* (reviewed in *Aggleton et al., 2011*, and *Carlesimo et al., 2015*), and few studies have so far reported evidence in favour of the MD's role in familiarity. Consequently, it has been suggested that the MD plays an indirect role in recollection because of its pattern of connectivity with the frontal lobes. A distinction has been drawn between the MDpc, which may be involved in recollection, and the MDmc, whose role remains more elusive (*Pergola et al., 2012*; *Carlesimo et al., 2015*). Furthermore, the role of other thalamic nuclei, such as the midline and intralaminar nuclei, which are often damaged along with the MD, has also been discussed.

Since neuropsychological investigations have pointed to a mixed pattern rather than a pure dissociation, *Aggleton et al. (2011)* revised their initial dual-process model of recollection and familiarity to integrate the complex connectivity of the thalamus. Their new multi-effect multi-nuclei (MEMN) model took into account the specific connectivity pattern of each thalamic nucleus. It described a continuum, rather than a dissociation, between the MTT/AN and MD. Furthermore, it suggested that the midline and intralaminar nuclei play a transitional role in recollection and familiarity (i.e., they influence these processes to varying extents). The authors particularly emphasized the MD's role in familiarity, owing to its afferent connection from the perirhinal cortex, as well as in other cognitive functions, which could then impact recollection.

Overall, Aggleton's models have received only mixed support concerning the role of the MD in familiarity, and this nuclei's more general role in memory remains to be clarified. One of the problems facing researchers is the difficulty of recruiting large homogeneous groups of patients. Many studies report on one or a few patients at the most, and when samples are larger, they often include patients with both right and left damage to the thalamus, even though the thalamus exhibits a laterality effect (*Edelstyn et al., 2012*). In addition, the methods used to identify which thalamic nucleus has been damaged are usually limited to visual inspection, or else do not take all the damaged

nuclei into account. Consequently, the aim of the current study was to overcome these limitations and assess how familiarity and recollection are affected by thalamic stroke, depending on which nuclei or bundles (e.g., MTT) are damaged. For this purpose, we recruited 14 patients with a first left thalamic stroke, along with 25 matched controls. All participants underwent a series of three verbal recognition memory tasks, each measuring recollection and familiarity in a different way, thus allowing us to assess these processes independently of the method used (*Yonelinas, 2001*; *Bowles et al., 2007*). An automated atlas was used to identify the location and extent of the damage to thalamus nuclei on the patients' high-resolution 3D MRI (*Danet et al., 2015*). Two complementary methods were used to assess damage to the MTT. Given the updated MEMN model, we expected to observe impaired recollection in the case of AN or MTT lesions, and impaired familiarity and recollection in the case of MD lesions (*Aggleton et al., 2011*).

## Results

### Participants

We recruited 14 patients with a left ischemic thalamic lesion in the stroke units of the university hospitals of Toulouse and Bordeaux (France). Our recruitment criterion was the detection of a first symptomatic thalamic infarct, regardless of initial symptoms or neurobehavioural report at onset. Only left thalamic strokes were included, in order to ensure a homogenous group. Patients were included at least 3 months after their stroke, had no history of previous neurovascular, inflammatory or neurodegenerative diseases, and had to be right-handed or ambidextrous. We excluded one patient because of a depressive syndrome that impacted cognition, and one patient because a lacunar lesion was only visible on the T2 sequence in the acute phase. The final sample therefore contained 12 patients (P1 to P12) along with 25 healthy participants matched for age and education (*Table 1* for demographic data of both groups; see lesions on structural MRI scans in axial view in *Figure 1* and in coronal view in *Figure 1—figure supplement 1*). All the participants underwent a standard neurological examination, a standard neuropsychological assessment, three verbal recognition memory tasks, and a high-resolution 3D MR scan. We carried out all the investigations in a single day and in the same order.

### Standard neuropsychological assessment

The participants underwent a comprehensive cognitive assessment. Patients performed less well than controls on verbal memory tasks (p<0.01 for all variables), and their executive functions and language were moderately impaired (*Table 2*, *Supplementary file 1*). No significant difference was found between patients and controls on the visual memory tasks although the recall of the Rey figure tended to be impaired, and behavioural assessments.

**Table 1.** Mean (standard deviation) [min, max] demographic data of patients and controls, and patients in the dMTT and iMTT subgroups.

MannWhitney and $\chi^2$ tests were used to compare patients and controls, and permutations tests and $\chi^2$ to compare dMTT and iMTT.

| | Left thalamic infarct patients (n = 12) | Healthy control participants (n = 25) | P value | dMTT subgroup (n = 7) | iMTT subgroup (n = 5) | P value |
|---|---|---|---|---|---|---|
| Age (years) | 53.2 (14.6) [25, 75] | 52.6 (11.6) [25, 69] | 0.86 | 58.9 (16.6) [25, 75] | 45.2 (6.3) [38, 52] | 0.12 |
| Sex (female (F)/male (M)) | 3F/9M | 15F/10M | 0.05 | 1F/6M | 2F/3M | 0.31 |
| Education level (years) | 12.8 (4.1) [5, 17] | 13.6 (4.1) [5, 21] | 0.25 | 12.3 (4.2) [5, 17] | 11 (4.2) [5, 17] | 0.69 |
| Handedness (right (R)/left (L)/ ambidextrous (A)) | 11 R/1A | 22R/3L | 0.17 | 6 R/1A | 5 R | 0.38 |
| Time since onset | 589 (588.9) days [3 months, 4 years 11 months] | – | – | 527 (647.2) days [3 months, 4 years 11 months] | 675 (556.1) days [3 months, 3 years 8 months] | 0.69 |
| Normalized volume of overall lesions (mm$^3$) | 516.8 (265.2) [30, 982] | – | – | 679.6 (160.7) [538, 982] | 289 (208.5) [30, 605] | 0.005 |

DOI: https://doi.org/10.7554/eLife.28141.004

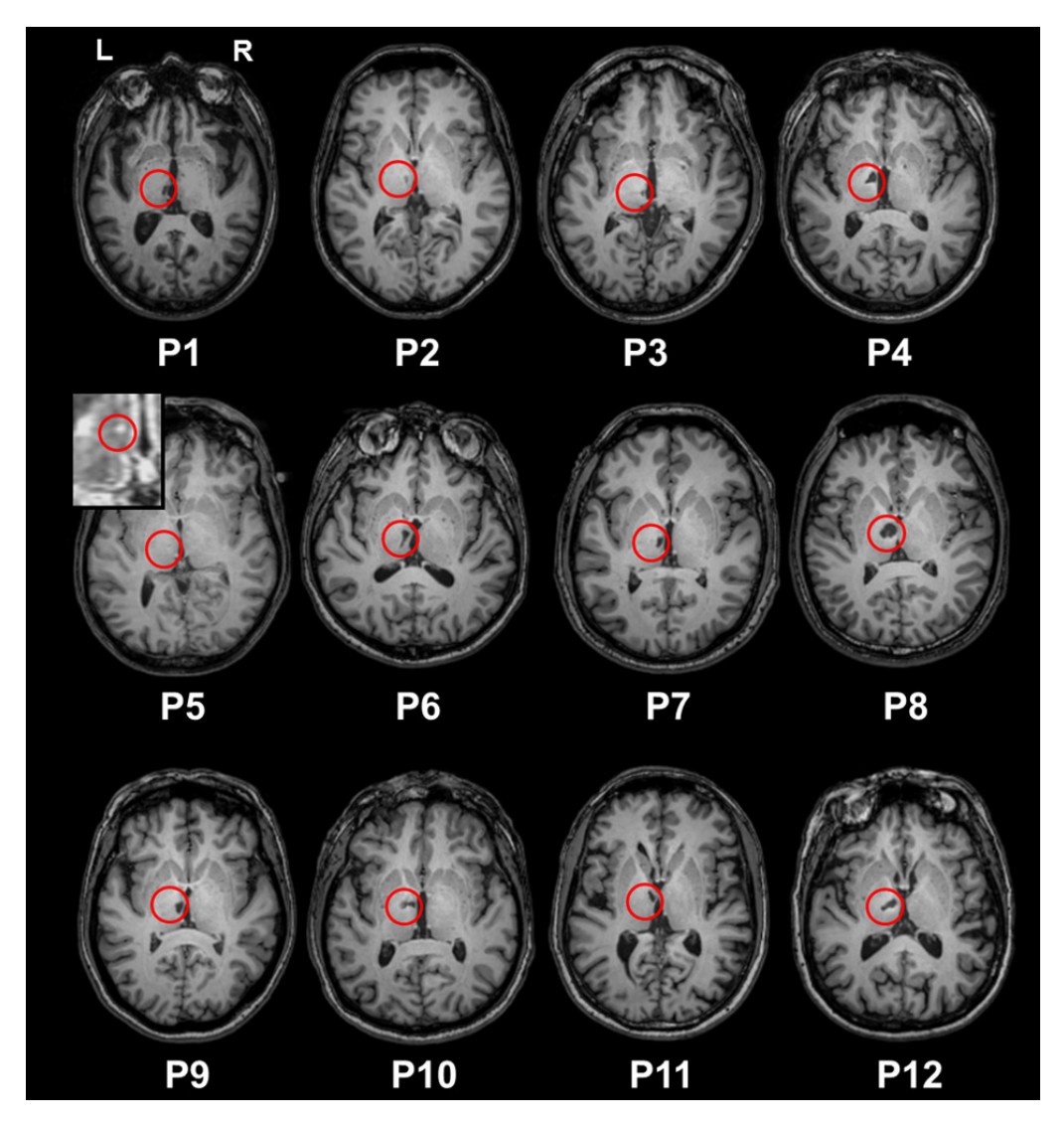

**Figure 1.** T1 axial sections of the patients' native brains. The red circles indicate infarcts. P5's lesion is hardly visible on the picture (lesion volume = 5 mm$^3$). We therefore provide a zoom on the Flair image, where the lesion is easier to see.

DOI: https://doi.org/10.7554/eLife.28141.002

The following figure supplement is available for figure 1:

**Figure supplement 1.** T1 coronal sections of the patients' native brains.

DOI: https://doi.org/10.7554/eLife.28141.003

## Recognition memory tasks

We used three different verbal recognition memory tasks to measure recollection and familiarity, each relying on a different procedure, in order to obtain recollection and familiarity estimates that were not dependent upon a specific task or estimation procedure (*Yonelinas, 2001*; *Bowles et al., 2007*; *Diana et al., 2007*). *Figure 2* (see *Supplementary file 1* for the details) shows the results of patients and controls for the three recognition indices d' (global performance), R (Recollection), and F (Familiarity) for each of the three experimental tasks: Receiving Operating Characteristics (ROC), Process Dissociation Procedure (PDP), Remember-Know-Guess paradigm (RKG). These results were highly convergent: patients' discrimination and recollection were impaired in all three tasks after correction for multiple comparisons, whereas familiarity was preserved (d' comparison between patients and controls: ROC $U = 42.5$, p=0.001, A = 0.86; PDP $U = 72.5$, p=0.04, A = 0.76; RKG $U = 23$,

**Table 2.** Median [min, max] results of the standard neuropsychological assessment. MannWhitney tests were used to compare the groups.

| Tasks | Subtests | Patients N = 12 | Controls N = 25 | P value |
|---|---|---|---|---|
| MEMORY | | | | |
| FCSRT - verbal | - Delayed total recall/16 | 10.0 (1, 16) | 16.0 (15, 16) | <0.01 |
| | - Recognition/48 | 44.5 (5, 48) | 48.0 (47, 48) | <0.01 |
| Logical memory - verbal | - Delayed recall (30 min)/50 | 16.5 (3, 37) | 38.0 (24, 46) | <0.0001 |
| | - Recognition/30 | 24.0 (16, 28) | 28.0 (23, 30) | <0.01 |
| Rey figure - visual | - Delayed recall (2 min)/36 | 19.8 (3, 32) | 27.0 (17, 34) | 0.053 |
| DMS 48 - visual | -Delayed forced-choice recognition (60 min)/48 | 47.5 (44, 48) | 47.0 (38, 48) | 0.36 |
| EXECUTIVE FUNCTIONS | | | | |
| Auditory-verbal span - Ss | | 8.0 (4, 14) | 13.0 (9, 18) | <0.01 |
| Visuospatial span ($n = 11$) - Ss | | 11.0 (5, 16) | 13.0 (9, 19) | <0.05 |
| Digit symbol - Ss | | 9.5 (5, 12) | 12.0 (8, 18) | <0.01 |
| Stroop | - Errors | 0 [0, 6] | 0 [0, 4] | 0.33 |
| Literal fluency (p) | - Number of words in 2 min | 15.5 (8, 23) | 26.0 (11, 42) | <0.0001 |
| Semantic fluency (animals) | - Number of words in 2 min | 22.5 (16, 40) | 42.0 (32, 61) | <0.0001 |
| LANGUAGE | | | | |
| Confrontation naming/36 | | 33.5 (26, 36) | 36.0 (35, 36) | <0.001 |
| BEHAVIOUR | | | | |
| Starkstein Apathy Scale/42 | | 9.5 [0, 18] | 8.0 (1, 19) | 0.33 |
| Beck Depression Inventory | | 3.0 [0, 8] | 2.0 [0, 13] | 0.55 |
| State-trait anxiety/80 ($n = 11$) | | 38.0 (28, 51) | 40.0 (23, 57) | 0.69 |

Note. Ss = scaled score. n = 11 indicates that one of the patient did not undergo this task.
DOI: https://doi.org/10.7554/eLife.28141.005

p<0.001, A = 0.92; recollection comparison ROC $U$ = 37.5, p<0.001, A = 0.88; PDP $U$ = 71.5, p=0.047, A = 0.76; RKG $U$ = 25.5, p<0.0001, A = 0.92; Familiarity comparison: ROC U = 104, p=0.1; PDP U = 92.5, p=0.06; RKG U = 114, p=0.3). We, therefore, computed summary scores across the three tasks (last row of **Figure 2**, mean $z$ scores across ROC, PDP and RKG tasks). $zd'$ and $zR$ were evidently lower in patients ($U$ = 23, p<0.00001, A = 0.92 and $U$ = 11, p=0.00001, A = 0.96). Familiarity was also found to be impaired ($U$ = 80, p=0.02, A = 0.73). Recollection correlated with global performance (rho = 0.65, p=0.05), but familiarity did not. The response criteria were not different between patients and controls in the ROC (ROC *c in patients: median* = −0.3, *min* = −1.4, *max* = 0.6; $U$ = 124, p=0.4) and RKG tasks (RKG *c in patients: median* = −0.4, *min* = −1.5, *max* = 1; $U$ = 110, p=0.2) whereas in the PDP task the patients' bias was significantly more conservative (PDP *c median* = −0.3, *min* = −1.4, *max* = 0.8; $U$ = 61.5, p<0.01).

Although group results indicated impaired recollection and a modest impairment of familiarity, there was a possibility that individual patients might have displayed different patterns (e.g., impaired familiarity and preserved recollection, or *vice versa*). To check this, we calculated the correlations between the $zF$ and $zR$ indices (**Figure 3**). As shown, there was a strong correlation in the patient group (rho = 0.85, p=0.05). Furthermore, none of the patients showed a tendency to be an outlier. No such correlation was found in the control group. We therefore also failed to find a dissociation between recollection and familiarity at the individual patient level.

Lastly, we failed to find any correlations between performances on the executive tests and $zd'$, $zR$ and $zF$. Even though we assessed correlations both with individual tests of executive functions, and by calculating summary scores across all the tests, as we did across the three recognition memory tasks.

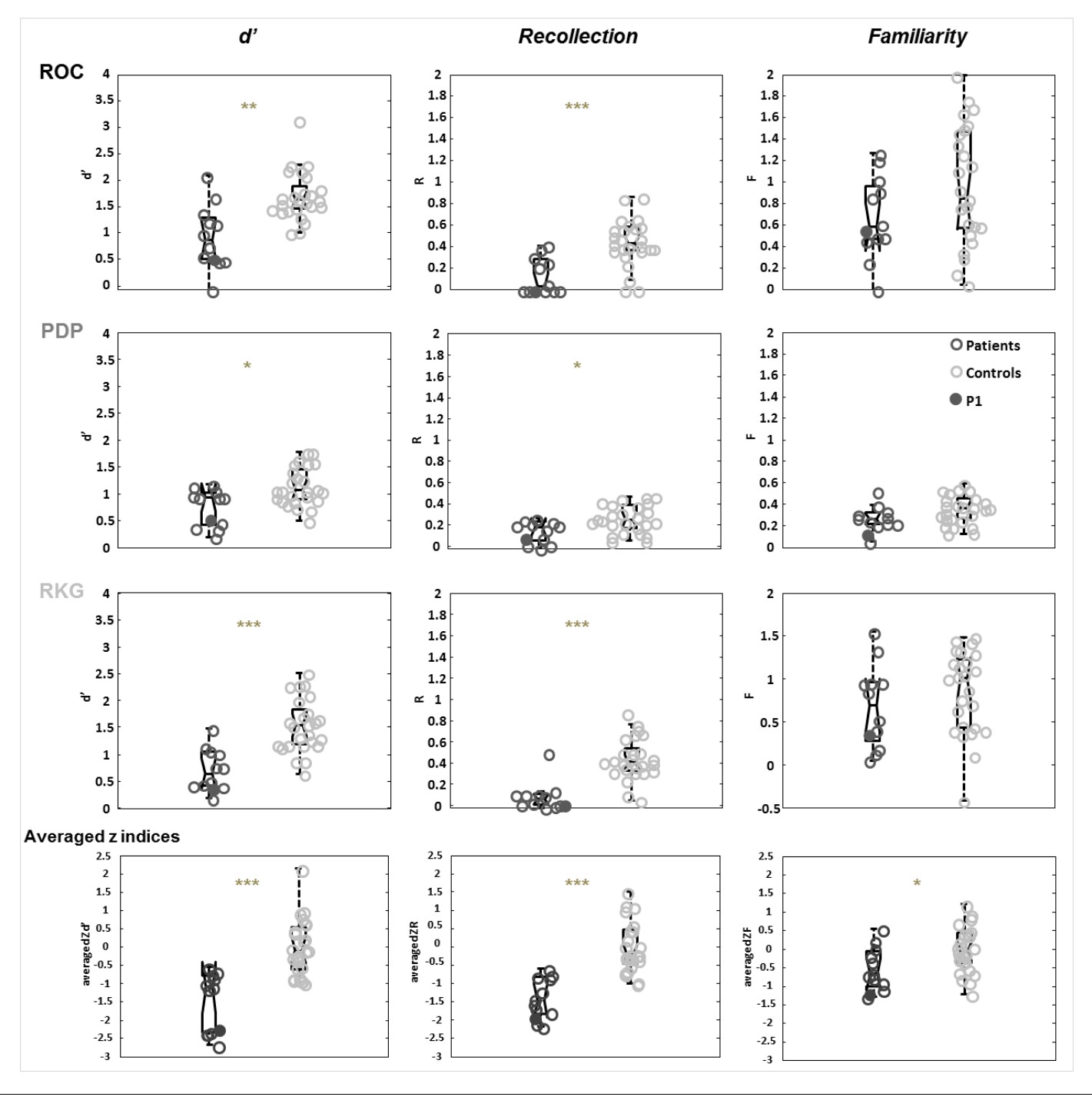

**Figure 2.** Comparison of patients and controls on the recognition memory tasks. Box plots represent the distribution in quartiles of the d', R and F indices for the ROC, PDP, RKG tasks, and for the summary scores across the three tasks (averaged z indices). Boxes represent the 25th and 75th percentiles, the lines in the boxes the medians. Notches display the variability of the median between samples. Boxplots whose notches do not overlap have different medians at the 5% significance level based on a normal distribution assumption. Comparisons are reasonably robust for other distributions, however, and statistical comparisons reported in the text were carried out independently of this graphical representation. Upper and lower lines of whiskers represent minimum and maximum performance. Outliers (i.e., subjects whose performance fall outside minimum or maximum values of 1.5 the difference between the 25th and 75th percentile) would be represented by circles outside the minimum and maximum values. Filled dark dots represent the case P1 whose MTT is intact according to the Morel atlas and damaged as stated in the volume analysis.

DOI: https://doi.org/10.7554/eLife.28141.006

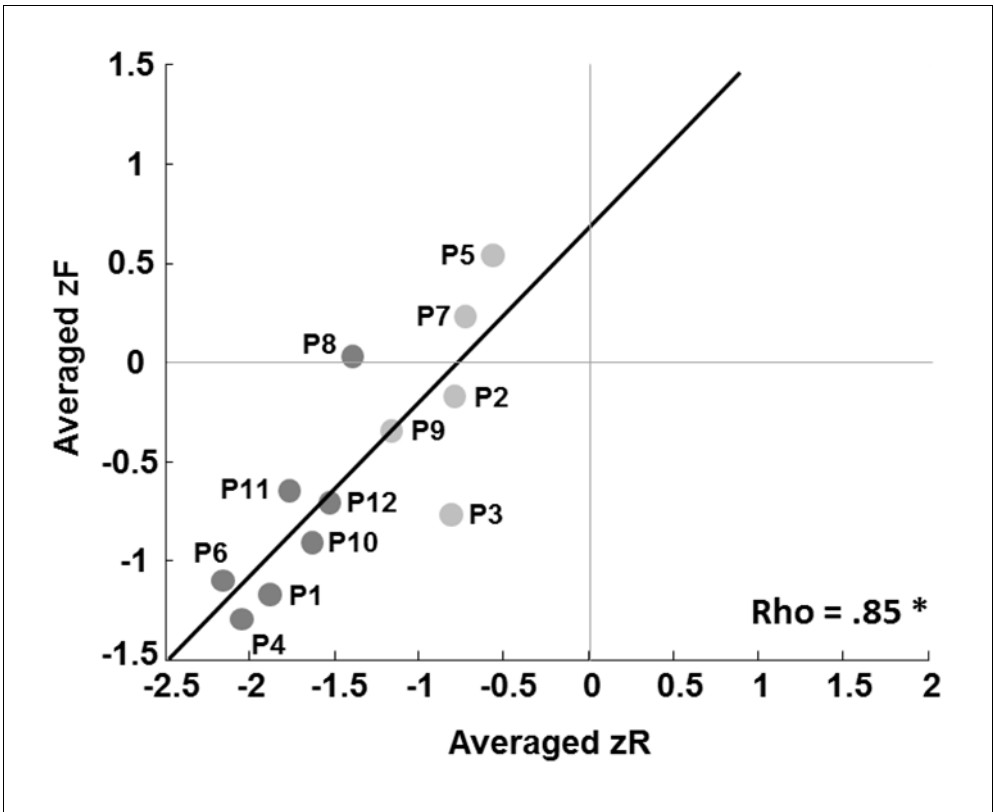

**Figure 3.** Correlation between patients' averaged zR and zF indices. Dark dots represent patients with a damaged MTT, and light dots patients with an intact MTT. The patient labels next to the dots correspond to those in *Supplementary file 2*, which details damage to the thalamic nuclei.

DOI: https://doi.org/10.7554/eLife.28141.007

## Volumetry and lesion localization

Only P1, P3 and P10 had lesions outside the thalamus (in the brain stem, red nucleus or white matter), and none of these involved brain areas known to play a role in declarative memory. The Fazekas and Schmidt score, which assesses white matter lesions, was $\leq 2$ for all patients and controls (*Kapeller et al., 2003*). Patients had lesions in the left medial group ($n = 11$), especially the MDpc, the intralaminar nuclei ($n = 12$), and the midline nuclei ($n = 11$) (*Figure 4*; details for individual patients in *Supplementary file 2*). Lesions were observed in the lateral group for 9 patients. As can be seen in *Supplementary file 2*, the extent to which these various nuclei were damaged varied greatly from one patient to another. It is noteworthy that only one patient had a very minor damage in the anterior group (1 mm$^3$ in the AN), and only one had a very small lesion in the posterior group (1 mm$^3$ in the limitans nucleus). Thus, with regard to Aggleton et al.'s models, none of the patients had a significant or isolated AN lesion, while 11/12 had MD lesions.

No correlations were found with the executive tests, nor between recognition indices (zd', zR and zF) and the total volume of the lesion. No correlation of these indices with the volume of the MDpc or MDmc was found either.

MTT volumetric analysis revealed atrophy of the MTT in seven patients. In six of these, the MTT was also labelled *damaged* using Morel's atlas, confirming the convergence between the two assessment methods. We included all seven patients in the damaged MTT subgroup, and the other five patients in the intact MTT subgroup. Thus, in line with Aggleton *et al.*'s models, damaged MTT patients had a lesion of the AN/MTT complex, while intact MTT patients had an MD lesion (except for P3), as well as varying degrees of damage to the intralaminar and midline nuclei. The two groups were not different in age and scholarship level (*Table 1*).

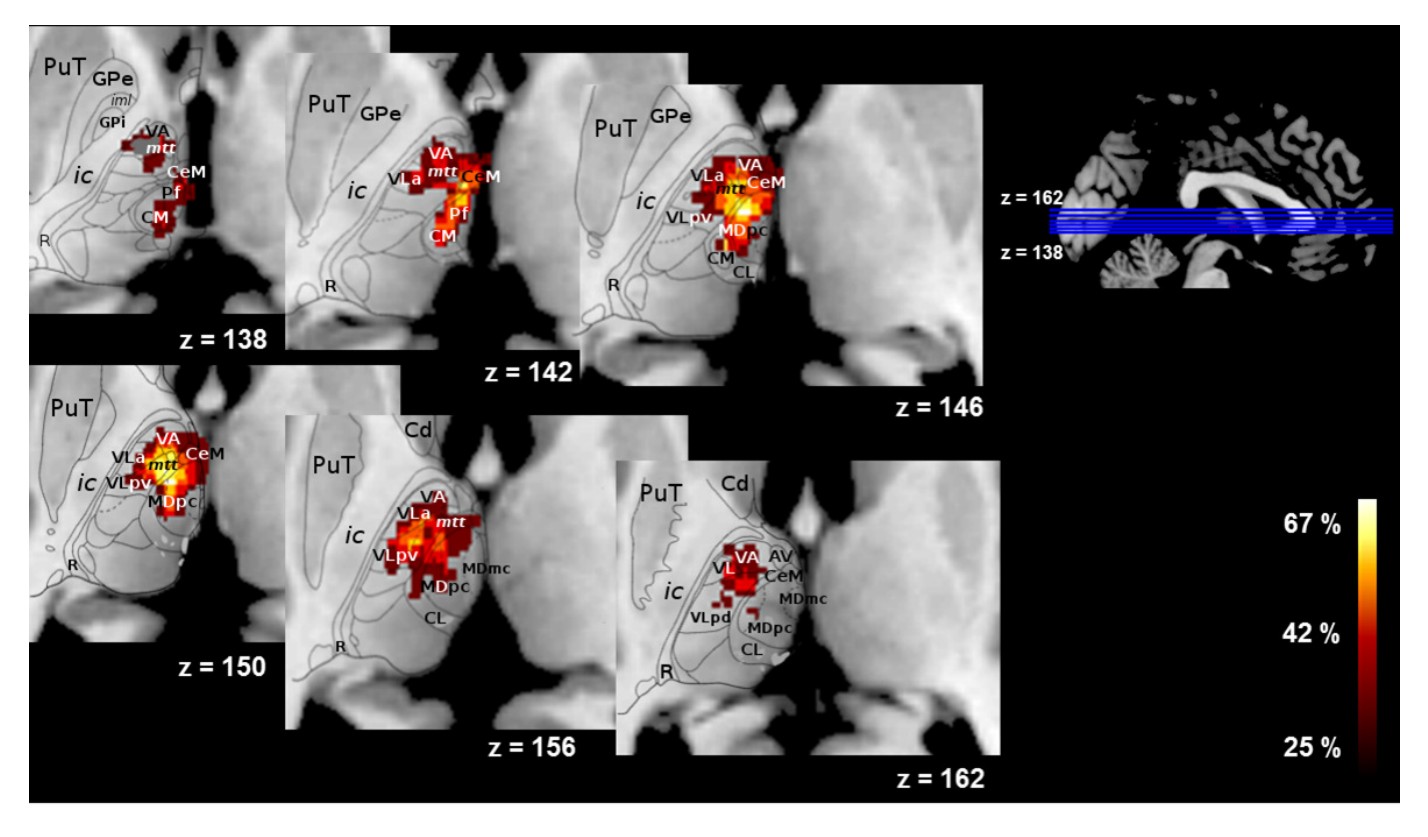

**Figure 4.** Overlap of the lesions across patients (% of patients, N = 12) on an axial view on the automated Morel atlas. PuT = putamen; GPe = external globus pallidus; ic = internal capsule; R = reticular nucleus; VA = ventral anterior; mtt = mammillothalamic tract; CeM = central medial; CM = central median; CL = central lateral; Hb = habenula = MD=mediodorsal.

DOI: https://doi.org/10.7554/eLife.28141.008

## Subgroup comparisons

The damaged MTT subgroup had a poorer mean performance ($zd'$: Z = −2.07, p=0.049), and displayed poorer recollection ($zR$: Z = −2.98, p=0.001) and familiarity ($zF$: Z = −2.11, p=0.03) than the intact MTT subgroup (*Figure 3* and *Figure 5*, *supplementary file 2*). The intact MTT subgroup had a lower $zd'$ and a lower $zR$, but their $zF$ was similar to that of controls ($zd'$: Z = −2.84, p<0.01, and $zR$: Z = −2.22, p<0.05).

We had previously compared the performance of the damaged and intact MTT subgroups to a standard verbal memory task. This study showed more severe impairment of both recall and recognition of the damaged MTT subgroup (*Danet et al., 2015*).

## Discussion

In the present study, we found that a large group of patients with left thalamic infarcts involving mainly the MD nucleus showed impaired recollection. Among the patients, those with MTT damage exhibited lower recognition performance. Unexpectedly, and contrary to the prediction that could be made following current models, patients with MD damage and intact MTT showed no familiarity impairment as well.

At first sight, our results appear to contradict the predictions made by Aggleton and colleagues models. In the original model, the MD supported familiarity (*Aggleton and Brown, 1999*), and the AN recollection. In the revised model, the MD supports familiarity and has an indirect effect on recollection. Both models predict that familiarity will be impaired following an MD lesion, regardless of whether recollection is impaired. *Aggleton et al. (2011)* stressed that this hypothesis remained

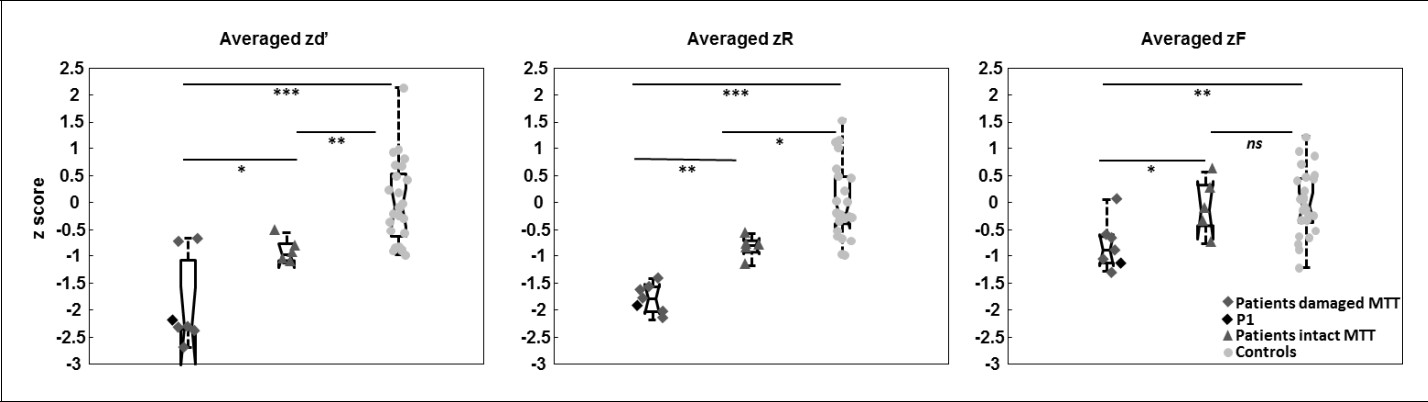

**Figure 5.** Comparisons of averaged z recognition indices (d', R, F) between the damaged MTT and intact MTT subgroups and controls, using permutation tests. *p<0.05. **p<0.01. ***p<0.001. ns = non significant. Boxes represent the 25th and 75th percentiles, the lines in the boxes the medians. Notches display the variability of the median between samples (Same details than described in the legend of the *Figure 2*). The black diamond represents the case P1, whose MTT is intact according to the Morel atlas but damaged as found in the volume analysis.
DOI: https://doi.org/10.7554/eLife.28141.009

unproven, even though two previous studies had suggested that there was no clear evidence of impaired familiarity following MD lesions (*Zoppelt et al., 2003*; *Soei et al., 2008*). However, a recent single-case study did report impaired familiarity following MD damage (*Edelstyn et al., 2016*). Using a task based on subjective report (Remember-Know paradigm), the authors found a dissociation between recollection and familiarity in patient OG. In a previous publication, they had localized the patient's damage in the right MD, internal medullary lamina, intralaminar and midline nuclei according to *Mai et al. (2004)* and the method of *Carlesimo et al. (2007)*. Comparing OG with 10 matched healthy controls they observed a significantly lower familiarity only for visual material (faces) whereas recollection was spared in the verbal and visual modality. In our study, we did not find any impairment of familiarity following an MD lesion in our group of 12 patients. We assessed familiarity in three different tasks, but none of these revealed a significant impairment, either at group or individual level. This was all the more surprising as our patients had a wide variety of thalamic lesions, although all of these were focused on the MD region. We had, therefore, expected at least some of the patients to display impaired familiarity. We found a moderate impairment of familiarity after averaging the familiarity indices across the three tasks (effect size A = 0.73). This appeared to be explained by lower overall familiarity in patients with concomitant MTT damage (*Figure 5*). Several possibilities need to be considered, however, before we can reach a conclusion as to the meaning of this finding.

A possibility is that familiarity is not impaired following MD lesions. Familiarity could in fact not depend on the MD, and possibly not on any thalamic nucleus at all. Familiarity is assumed to be a fast process (*Brown and Aggleton, 2001*), and it could be argued that direct connections between the MTT and the prefrontal cortex are more efficient than connections relayed by the thalamus although there are neural mechanisms that keep thalamo-cortical conduction velocity constant and fast across the cortex (*Salami et al., 2003*).

Interestingly, there are no connections between the perirhinal cortex and the thalamus in rodents (although, there are in nonhuman primates; for a review, see *Aggleton et al., 2011*), suggesting that this thalamic relay may not be absolutely critical in recognition memory tasks. Both lesion and electrophysiological studies in non-human primates suggested that the MD might play a role in recognition memory (*Aggleton and Mishkin, 1983a*, *1983b*; *Parker et al., 1997*; *Fahy et al., 1993*). However, the impact on the performance of lesions of the MD was moderate (i.e., less severe than direct perirhinal cortex lesions). Furthermore, the idea that the MD could play a role in familiarity was not demonstrated, but seems to be merely an inference stemming from the observation that there are some direct connections from the perirhinal cortex to this nucleus (*Aggleton et al., 1986*; *Russchen et al., 1987*; *Aggleton and Mishkin, 1983a*). However, the importance of these connections and their functional role remains to be clarified. For example, the activity of the AN and of the

neocortex was registered using implanted electrodes in epileptic patients during a memory task. The authors found that the activity of the two regions became synchronized during successful storage, providing direct evidence of the involvement of the AN in memory (*Sweeney-Reed et al., 2014*). Such a direct measure of the activity of the MD during a task based on familiarity could help making progress on this issue.

The idea that familiarity could be selectively impaired is based on a dual view of familiarity and recollection, whereby these processes would be functionally and anatomically independent. Many studies have shown that recollection can, indeed, be selectively impaired (*Tsivilis et al., 2008*; *Vann et al., 2009*). Recollection, therefore, appears to depend on a relatively well-circumscribed neural network, hierarchically organized, in which the hippocampus and diencephalic structures are critical components. Any lesion to this network impairs recollection, particularly since some of these areas are particularly sensitive to various neurological insults and are rather small (and thus easily damaged in their entirety). It is, therefore, tempting to see familiarity as a process paralleling recollection, both functionally and anatomically, with a similar network of dedicated brain areas. However, this does not have to be necessarily the case. Although it is quite easy to find patients with severe isolated impaired recollection, finding patients with isolated impaired familiarity remains surprisingly difficult to evidence. Indeed, very few studies have reported impaired familiarity but preserved recollection (*Bowles et al., 2007*; *Martin et al., 2011*; *Brandt et al., 2016*). Therefore, a possibility could be that familiarity could depend on a wider, more diffuse and partly redundant neural system. For example, the areas processing familiarity in the visual ventral streams could be rather large so that after a lesion, remaining preserved cortical patches could still partly process familiarity. Following up on this idea, it could be that the MD plays a role in familiarity, but that the neural system supporting familiarity could cope with MD lesions through redundancy or direct temporo-frontal connections. Here, we argue that the models of the brain network supporting familiarity could be revised without a priori attempt to parallel the one supporting recollection.

This idea is supported by recent suggestions that the view of familiarity as a single process is oversimplified, and that it actually follows a cascade of different simpler processes, such as perceptual and conceptual fluency, process attribution and post-retrieval monitoring (*Whittlesea and Williams, 2000*; *Montaldi and Mayes, 2010*; *Besson et al., 2015*). In other words, it is as yet unclear which aspect of familiarity is impaired after an MD lesion. If it is a higher-order process, performance could remain at a reasonably good level, but with impairment of some phenomenological aspect of familiarity. As recognition memory tasks are usually not designed to assess familiarity subprocesses, this may have gone unnoticed in both ours and previous studies. Future studies will therefore need to include tasks that concentrate on specific characteristics of familiarity, such as speed (*Besson et al., 2012*, *2015*) or visuoperceptual processing (*Migo et al., 2009*).

In sum, familiarity was not found to be impaired across three different tasks relying on different measures of familiarity following MD damage, apparently contradicting the simple, dual-process, view of the role of the thalamus in memory. By contrast, this finding is a call to revisit models of familiarity and the role the MD plays in this process.

This leaves open the question of why the patients with damage to both the MD and the MTT exhibited impaired familiarity, whereas those with MD damage alone did not. Interestingly, the only other patient to be described in the literature as displaying impaired familiarity following MD damage also had an MTT lesion (*Edelstyn et al., 2016*), thus supporting our findings. Consequently, combined MTT and MD lesions may impair familiarity. This would hold true only for some patients, since not all our patients with combined MTT/MD damage exhibited impaired familiarity, compared with controls. Only one patient in our study had a lower familiarity score than that of the poorest performing control. One explanation for this finding is that another tract was damaged along with the MTT in some patients, such as the inferior thalamic peduncle, which connects the perirhinal cortex to the MD (*Aggleton and Brown, 1999*). An alternative explanation, however, is that this was related to damage to other thalamic nuclei. The patients with combined MTT/MD damage also had larger thalamic lesions overall, involving other nuclei besides the MD, such as the midline nuclei. They also present lower performance on recognition memory tasks (*Figure 3*) or memory in general (*Danet et al., 2015*) so that a specific role for the MTT in familiarity seems at present improbable. Undetermined anatomical factors could explain this result.

Our group of patients had impaired recollection. Given the known role of the MTT/AN complex in recollection (*Tsivilis et al., 2008*; *Vann et al., 2009*), it is no surprise that patients with lesions to

this complex had impaired recollection. However, our subgroup of patients who had lesions in the MD region, but not of the MTT/AN complex, also exhibited impaired recollection. Recollection correlated with performance, but familiarity did not, suggesting that impaired recollection was responsible for impaired performance.

These findings appear to be highly consistent with previous results. *Pergola et al. (2012)* measured the contribution of recollection using a dissociation paradigm. Participants had to learn picture pairs. In a yes/no recognition phase, they were shown single-picture targets mixed with distractors. For all 'yes' responses, they were asked to recall the other picture in the pair. Twelve patients with MD lesions (six left, six right) were included in their study. Results showed that cued recall, taken as an index of recollection, correlated with MDpc volume loss. In line with *Van der Werf et al. (2003)*, Pergola *et al.*, therefore, argued for a role of this region in recollection, owing to its connectivity with the dorsolateral prefrontal cortex. More recently, *Tu et al. (2014)* reported selective impairment of delayed recall in seven patients with a left MD lesion but no MTT damage. Across these studies, the MD's role in recall and recollection appears clear. This is also the case in our study. These findings are globally in line with *Aggleton et al. (2011)*'s model, and appear to corroborate this part of it.

*Mitchell and Chakraborty (2013)* reviewed the findings on MD lesion effects in 52 animal studies. These authors found that the MD has a number of subdivisions, each with its own neural circuit connecting it with the prefrontal cortex. They suggested that the MD plays a broad role in the regulation of cortical synchrony between medial temporal lobe structures and the prefrontal cortex. According to this view, recollection could be impaired following MD damage because of this lack of synchrony. Actually another explanation as to why the MD could be involved in recollection is that it could be involved in executive functions. However, we did not find any such correlation in the present study. Of note, nuclei other than the MD (central median and parafascicular) have been associated with a dysexecutive syndrome (*Liebermann et al., 2013*).

It should be noted that more work needs to be done to better understand the involvement of the thalamus, and particularly of the MD, in memory. It is at present difficult to precisely image the nuclei and tracts running within the thalamus. Dedicated structural MRI sequences, rather than state of the art but standard ones, could be developed in the near future and help refining current results. 7 Tesla, as opposed to current 3 Tesla, imaging in patients could also potentially be useful. This is important since specific lesions of the MD in non-human primates may result in quite different lesions than those resulting from thalamic strokes in the human. Comparisons are thus not entirely straight forward. There are also currently debates regarding how recollection and familiarity should be quantified and modelled, and whether dual-process approaches such as the ones we used in the current study, although widely used, are appropriate (*Wixted et al., 2010*; *Pazzaglia et al., 2013*; *Moran and Goshen-Gottstein, 2015*; *Didi-Barnea et al., 2016*). More specific issues have also not been clearly addressed in the literature on the thalamus regarding, for example, the impact of a possible left/right thalamic asymmetry on *Aggleton et al. (2011)* model. For example, we used verbal stimuli (words) in patients with left-sided lesions of the thalamus. However, it is noteworthy that the only study having reported impaired familiarity in a patient with MD (and MTT) lesions used faces in a patient with right-sided lesion. As noted earlier, future studies in patients could also favour tasks focusing specifically on familiarity processes (*Migo et al., 2009*; *Besson et al., 2012*).

In conclusion, even if the role of the MD in recognition memory becomes clearer, work needs to be continued to clarify the involvement of the thalamus in memory. Our study suggests that models of familiarity assigning a critical role to the MD should be reappraised.

## Materials and methods

### Ethics and participants

All participants provided written informed consent in accordance with the declaration of Helsinki to take part in this study, which was approved by the local institutional review board (Comité de Protection des Personnes Sud-Ouest et Outre-Mer no. 2-11-04). Patients with single unilateral left ischemic thalamic stroke were recruited in the stroke units of Toulouse and Bordeaux university hospitals (France).

## Standard neuropsychological assessment

We tested *verbal memory* (Free and Cued Selective Reminding Test, *Van der Linden, 2004*); Logical Memory, *Wechsler, 2001*), *visual memory* (Rey-Osterrieth Complex Figure, *Rey and Vaivre-Douret, 1960*); DMS48, *Barbeau et al., 2004*. The latter is a clinical recognition memory test that was not included in the experimental analyses.), *executive functions* (Digit and Spatial Span, *Wechsler, 2001*; d2 test, *Brickenkamp and Zillmer (1998)*; Trail Making Test, *Godefroy and GREFEX, 2008*; Stroop test, *Godefroy and GREFEX, 2008*; Digit-Symbol test, *Wechsler, 1997*; literal and semantic lexical fluency, *Godefroy and GREFEX, 2008*; Similarities, *Wechsler, 1997*), *language* (ExaDé confrontation naming test, *Bachy-Langedock, 1989*), and *affects* (State-Trait Anxiety Inventory, *Spielberger, 1983*; Starkstein Apathy Scale, *Starkstein and Leentjens (2008)*; Beck Depression Inventory, *Beck and Steer, 1993*). Handedness was assessed with the Edinburgh Handedness Inventory (*Oldfield, 1971*).

## Recognition memory tasks

Each task was made of an encoding phase, a distractive phase of 10 min during which participants completed nonverbal tests and a recognition phase. The words were presented using Eprime v2. Participants typed their responses on a keyboard to monitor behaviour. For each task we computed three indices of interest: accuracy, computed as a d' reflecting the ability to discriminate between targets and distractors, and R and F indices. Accuracy was computed based on the signal detection theory, corrected according to *Snodgrass and Corwin (1988)*. R and F index calculation depended on each procedure, as described below. The response bias (conservative to liberal) was measured in each task and corresponds to the signal detection criterion (c corrected). Because there were three tests and because results were highly consistent across the three tasks, these indices were also averaged for each patient after a z-score transformation (using the control subjects mean and standard deviation) to obtain a summary score for each index.

The *ROC* task (*Figure 6A*) was adapted from *Yonelinas, 2001*. Participants incidentally encoded 120 concrete words presented sequentially. The words were concrete nouns presented in lowercase letters. The frequency of occurrence the words in printed texts (Lexique2.org, *New et al., 2001*) ranged from 0.5 to 241.7 (mean = 19.6, SD = 23.4). Words contained between 4 and 10 letters (mean = 6.5, SD = 1.1) and 1 and 3 syllables (mean = 2.0, SD = 0.5). Encoding was shallow for 60 words, (participants were told to press '1' if the number of syllables was less than two, and '2' if the number of syllables was equal or more than two) and deep for the other 60 words (participants had to rate the pleasantness of each word on a scale ranging from 1 (*very unpleasant*) to 7 (*very pleasant*)). After the 10 min interval, participants had to recognize the targets among distractors (*n* = 60) in a yes/no recognition task. For each response, they were asked to rate their confidence level on a 6-point scale ranging from 1 (*Sure it's new*) to 6 (*Sure it's old*). They were instructed to be as accurate as possible, but also to spread their answers across the scale, if possible (*Yonelinas et al., 1998*). Confidence-based ROC curves were generated for each participant and familiarity and recollection indexes were estimated using the Yonelinas High-Threshold model (*Yonelinas, 1994*; *Yonelinas et al., 1998*; for a review, see *Yonelinas and Parks, 2007*).

The *PDP* task (*Figure 6B*) was adapted from *Wolk et al. (2008)*. In the first phase, participants incidentally encoded 80 pairs of concrete words, half of them repeated three times. The words were concrete nouns presented in lowercase letters. The frequency of occurrence the words in printed texts (Lexique2.org, *New et al., 2001*) ranged from 1.6 to 199.4 (mean = 27.5, SD = 33.3). Words contained between 4 and 7 letters (mean = 5.7, SD = 1.0) and 1 and 3 syllables (mean = 1.7, SD = 0.5). To facilitate encoding, participants were asked to press '1' if the first word in the pair corresponded to the largest object, and '2' if it was the second word in the pair. After the 10 min interval, participants had to recognize target pairs (*n* = 40) among new pairs (both words new; *n* = 40) and recombined pairs (each word from a different pair at encoding; *n* = 40). They pressed '1' if the pair was old (target) or '2' if it was not a pair previously encoded (new/recombined). We derived familiarity and recollection indices followed the process dissociation procedure, as extensively reported in *Wolk et al. (2008)*. We included target pairs that had been correctly recognized, both recollection and familiarity may have helped recognition in this case. We excluded (incorrectly recognized) recombined pairs. In this case these responses were assumed to have been based on familiarity, since recollection would have prevented participants from endorsing them as old. We then

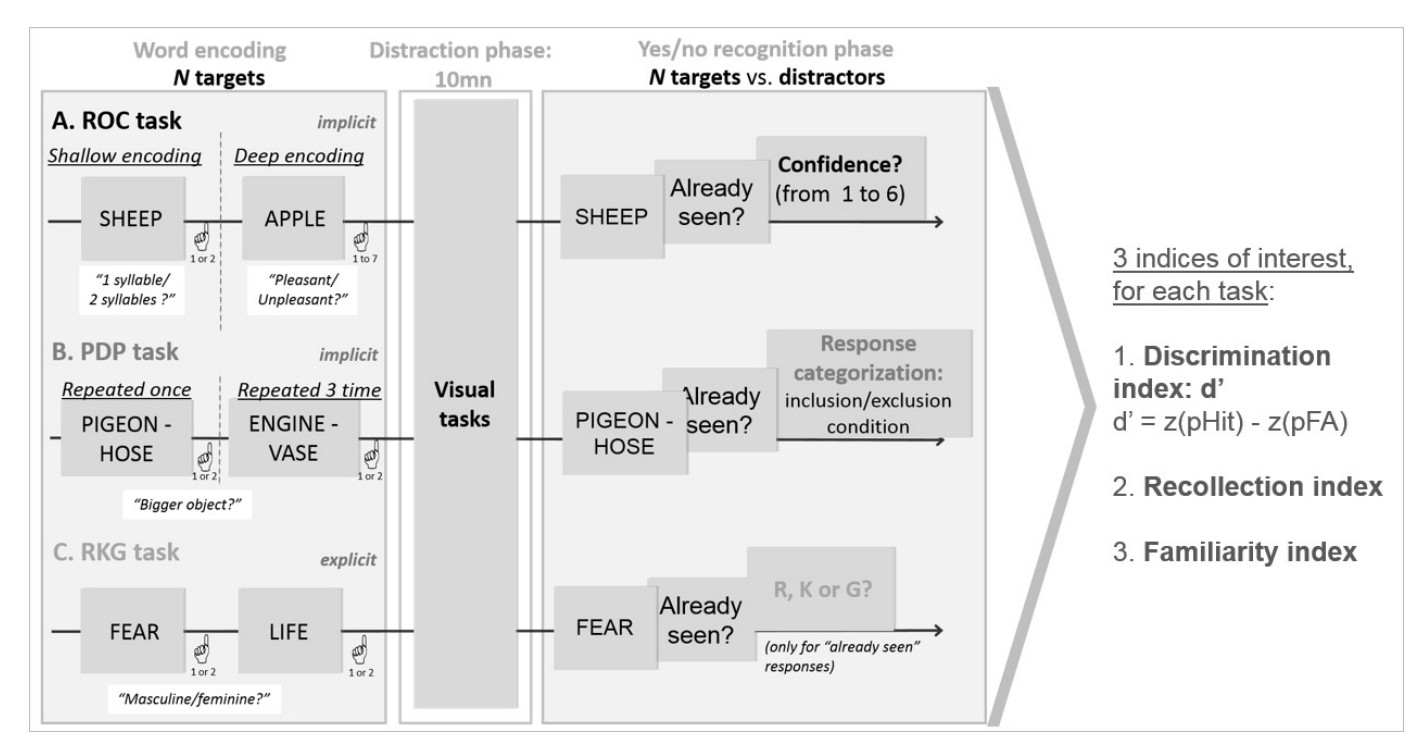

**Figure 6.** Experimental design of the three tasks (ROC, PDP, RKG). All verbal tasks consisted of an encoding phase, a distractive phase and a yes/no recognition phase. Supplementary questions in the ROC and RKG tasks allowed for the calculation of an index of global performance (d'), recollection and familiarity.

DOI: https://doi.org/10.7554/eLife.28141.010

subtracted included items from excluded ones (p(included) - p(excluded)) to calculate recollection scores, while familiarity score corresponded to the number of excluded items (p(excluded) / (1 R)).

The *RKG* task was based on Tulving's protocol (*Tulving, 1985*; *Gardiner, 2001*) (*Figure 6C*). Participants explicitly encoded 60 abstract words, of which 20 had a positive valence, 20 a negative valence, and 20 a neutral one categorized based on the results of an earlier pilot work. All were presented in uppercase letters. The frequency of occurrence the words in printed texts (Lexique2.org, *New et al., 2001*) ranged from 0.1 to 388.2 (mean = 32.6, SD = 56.6). Words contained between 3 and 13 letters (mean = 6.8, SD = 1.5) and 1 and 5 syllables (mean = 2.2, SD = 0.8). Participants pressed '1' if the word was masculine, and '2' if it was feminine during the encoding phase. After the 10 min interval, participants had to recognize the targets among distractors (n = 60) in a yes/no recognition task. For each 'yes' response, they were asked to say whether they remembered the item with reference to the encoding context (R responses), if they recognized the item without any context (K responses), or if they simply guessed (G responses). The probability of using recollection or familiarity was then estimated following *Yonelinas et al. (1998)*. The recollection index corresponded to the correct 'Yes' responses corrected for false alarms and divided by the probability for the response to be a R response. The familiarity index corresponded to the difference between old and new items distributions, measured using d'. None of the words was repeated across the tasks.

## Structural MRI acquisition and analysis

A 3T scanner was used to acquire MRI images (Philips Achieva). A three-dimensional T2-weighted sequence (1*1*1 mm voxel size, echo time = 337 ms, repetition time = 8000 ms, inversion time = 2400 ms, field of view = 240*240*170, slice thickness = 1 mm, slice number = 170) and a three-dimensional T1-weighted sequence (1*1*1 mm voxel size, echo time = 8.1 ms, repetition time = 3.7 ms, flip angle = 8°, field of view = 240*240*170, slice thickness = 1 mm, slice number = 170) were used to quantify the lesions. White-matter lesions were quantified with the Fazekas

and Schmidt score by two independent raters (LD and MP, modified kappa, $\kappa$ = 0.8) (*Kapeller et al., 2003*).

## Lesion volumetry

Two independent investigators (LD and PE) manually segmented the lesions on the native T1 images using MRIcron software (modified kappa, $\kappa$ = 0.82) (*Rorden et al., 2007*). After the native images and lesions had been normalized to the MNI (Montreal Neurological Institute) template (FSL), volumes expressed in $mm^3$ were automatically calculated for each patient (Fsl.anat toolbox).

## Lesion localization

Lesions were automatically localized using Krauth's digital version of Morel's atlas of the thalamus (FSL Atlasquery) (*Morel, 2007*; *Krauth et al., 2010*). We then measured the volume of the normalized lesions in each nucleus ($mm^3$) for each participant, as well as the proportion of lesions for each nucleus, using the labelled volumes of Krauth's version of Morel's atlas. We assessed the proportion of lesions outside the thalamus (expressed in %) (FIRST model-based sub-cortical structure segmentation tool, FSL).

## MTT assessment

An MTT label was included in Morel's atlas. Furthermore, we manually segmented patients' and controls' MTTs, and carried out a volumetric analysis using MRIcron software (two independent investigators, LD and PE). Patients were included in the damaged MTT subgroup if at least one of the two methods indicated damage. Segmentations with an inter-rater agreement below 70% were reviewed by the two raters together. Details about the lesion volumetry and localization, as well as the MTT assessment, are reported in *Danet et al. (2015)*.

## Statistical analysis

Analyses were carried out using $\chi^2$ for nominal data. We used the nonparametric MannWhitney $U$ test for comparisons between patients and controls, but opted for a permutation test, a procedure suitable for small sample size, to compare the performances of the dMTT and iMTT subgroups (*Ernst, 2004*).

Analyses were carried out with Statistica Version 8 and the *coin* (Conditional Inference Procedures in a Permutation Test Framework) package in R Version 3.0.3. Spearman's rho was used for nonparametric correlations. The level of significance was set at p=0.05. For the correlation analyses, d', R and F indices were averaged after they had been *z*-transformed according to the controls' means and standard deviations. We computed a nonparametric effect size based on ranks (*Vargha and Delaney, 2000*). This effect size can range from 0.5 to 1, with an A (measure of stochastic superiority) of between. 56 and. 64 corresponding to a small effect, one between 0.65 and 0.71 to a medium effect and one above. 71 to a large effect (equivalent values for Cohen's *d* of 0.2, 0.5 and 0.8). Multiple comparisons were corrected using the Bonferroni-Holm correction (*Holm, 1979*).

# Acknowledgements

We are grateful to Gabor Székely for sharing his digital atlas of the human thalamus. Jonathan Curot, Aicha Lyoubi and Marie Lafuma performed the neurological examinations. Hélène Gros, Nathalie Vayssière, Lucette Foltier and Jean-Pierre Désirat from the MRI technical platform (INSERM U1214) provided invaluable assistance. Finally, we thank all the participants for their motivation and enthusiasm.

# Additional information

## Funding

| Funder | Grant reference number | Author |
| --- | --- | --- |
| Toulouse Teaching Hospital | Local funding hospital grant | Jérémie Pariente |

The funders had no role in study design, data collection and interpretation, or the decision to submit the work for publication.

## Author contributions

Lola Danet, Conceptualization, Data curation, Software, Formal analysis, Supervision, Validation, Visualization, Methodology, Writing—original draft, Writing—review and editing; Jérémie Pariente, Conceptualization, Data curation, Software, Formal analysis, Supervision, Funding acquisition, Validation, Investigation, Visualization, Methodology, Writing—original draft, Project administration, Writing—review and editing; Pierre Eustache, Nicolas Raposo, Igor Sibon, Jean-François Albucher, Fabrice Bonneville, Data curation, Writing—review and editing; Patrice Péran, Conceptualization, Data curation, Software, Formal analysis, Visualization, Methodology, Writing—review and editing; Emmanuel J Barbeau, Conceptualization, Software, Formal analysis, Supervision, Visualization, Methodology, Writing—original draft, Writing—review and editing

## Author ORCIDs

Lola Danet (iD) http://orcid.org/0000-0001-8507-6749
Emmanuel J Barbeau (iD) http://orcid.org/0000-0003-0836-3538

## Ethics

Human subjects: All participants provided written informed consent in accordance with the declaration of Helsinki to take part in this study, which was approved by the local institutional review board (Comité de Protection des Personnes Sud-Ouest et Outre-Mer no. 2-11-04).

## Decision letter and Author response

Decision letter https://doi.org/10.7554/eLife.28141.014
Author response https://doi.org/10.7554/eLife.28141.015

# Additional files

## Supplementary files

• Supplementary file 1. Raw data (recognition tasks and neuropsychological assessment) for all the patients and healthy controls.
DOI: https://doi.org/10.7554/eLife.28141.011

• Supplementary file 2. Patterns of lesions for both intact and damaged subgroups. The normalized volumes of the lesions are expressed in $mm^3$. The extent of the lesions within the main thalamic nucleus groups (medial, lateral, anterior, posterior), subgroups (mediodorsal, intralaminar, midline) and individual nuclei (magnocellular MD, MDpc) is expressed as a percentage of volume loss according to Morel's atlas. MTT volume loss is expressed as a percentage, according to Morel's atlas. MTT volume is expressed as a $z$ score compared with control participants.
DOI: https://doi.org/10.7554/eLife.28141.012

• Transparent reporting form
DOI: https://doi.org/10.7554/eLife.28141.013

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
