## [Decision Letter]

Thank you for submitting your article "Medial thalamic stroke and its impact on familiarity and recollection" for consideration by *eLife*. Your article has been reviewed by three peer reviewers, and the evaluation has been overseen by a Reviewing Editor and Timothy Behrens as the Senior Editor. The following individuals involved in review of your submission have agreed to reveal their identity: John P Aggleton (Reviewer #1); Chris M. Bird (Reviewer #3).

The reviewers have discussed the reviews with one another and the Reviewing Editor has drafted this decision to help you prepare a revised submission. We hope you will be able to submit the revised version within two months.

We feel that your manuscript may make a potentially important contribution to understanding the neural basis of recollective and familiarity-based memory. However, there are several issues of presentation and interpretation that need to be clarified, as noted in the edited reviews below.

Reviewer #1:

The question at the heart of this study is one that needs resolving. The neuropsychological approach is logical but inevitably poses interpretation problems based on precise lesion reconstructions. The authors have sensibly just looked at left thalamic strokes in order to examine the effects of MD and/or MTT damage. While the MTT results are a consistent loss of recollection with variable loss of familiarity (see Figure 3) the cases with MD but not MTT damage are notable for their sparing of familiarity – which seemingly contradicts predictions by Aggleton.

The authors have gone to much length to detail the size and extent of the lesions (e.g. Supplementary file 2) as well as the figures in the main text. (It would be helpful to better flag the rather confusing case P1 re damage to MTT.)

The nonhuman primate work that provided the initial notion that MD has an important role in recognition (assumed to be familiarity) is barely considered or even referenced (see work of Gaffan, Parker, Aggleton, Mishkin, Squire) – though a review by Mitchell is considered. Why does this matter? Well those studies examined extensive cell loss in MDmc while the strokes in the present cases are more distributed between MDpc, MDmc and intralaminar n. (This pattern is quite typical of these kinds of strokes).

It would have been most helpful if coronal sections had been provided, so as to see how the strokes undercut MD and extend into MDpc. (I am assuming this is the pattern based on previous studies, but it’s incredibly important to know). For similar reasons, it is important (but incredibly difficult) to assess the amount of tract damage within the thalamus. These comments are not critical of the study but highlight why it is so hard to be definitive.

The authors mention that the thalamic group were impaired on verbal memory – which rather sets up the unsurprising result that as a group they were poor on recollective recognition – how does this relate to the two subgroups?

Overall this is a well conducted study – my only concern is the difficulty of teasing apart the impact of the various nuclear/tract effects – which is a pervasive issue. The results do seemingly contradict some of the predictions made by Aggleton and Brown in 1999 (rather a straw man I am afraid) but they seem to align much closer to Aggleton et al. 2011 – with the 'intact' familiarity in the MTT intact group providing a new perspective, linked to the idea that familiarity reflects a much wider, diverse set of processes.

*Reviewer #2:*

The study examines the effects of medial thalamic lesions on recollection and familiarity. 12 patients with left hemisphere strokes and damage to the thalamus were compared to 25 controls on three different experiments measuring recollection and familiarity (i.e., ROC confidence, process dissociation, and remember/know studies). The lesion locations were quantified to assess the recent dual process thalamic models of Aggleton, which proposes that damage to the AN or MTT would led to selective recollection impairment whereas MD lesions would lead to reductions in both recollection and familiarity. The results are found to provide only partial support for the models and indicate that patients with thalamic damage that does not impact the MTT were impaired on recollection but not familiarity, whereas patients with damage that included the MTT showed impairments on both recollection and familiarity.

There has been considerable interest in determining the role of different thalamic regions to episodic memory in general, and more specifically in recollection and familiarity. Unfortunately, the human literature has depended almost entirely on individual case studies. The current study represents an important advance in that it includes a reasonably good sized group of patients with homogeneous etiologies. Moreover, the patient's lesions in the current study were carefully quantified which overcome some ambiguities inherent in some of the previous work. Moreover, the current study is well designed and included multiple replications using various different measurement methods. In addition, the results seem remarkably clear and consistent across the subjects. Finally, the results really do present some important challenges to the current thalamic models. So, I think the paper will make a very important contribution to the literature.

However, I felt that the theoretical implications of the results should be discussed in more detail. This seems critical given there was a clear difference in the different patient groups and the fact that these results are not well described by the existing models. So, on the surface it would seem that the MD is critical for recollection, and that it is damage to the MTT that is critical for familiarity. I guess it just strikes me as odd that this is the case, and it seem worth speculating as to why this would be true.

In the second paragraph of the subsection “Recognition memory tasks”, it is concluded that "we therefore failed to find a dissociation between recollection and familiarity in any of the patients." This does not seem to be the case, however. As I understood the analysis the patients without involvement of the MTT did exhibit a selective impairment in recollection that left familiarity unaffected. So, this is a dissociation, in the traditional sense.

Reviewer #3:

In this neuropsychological study the authors report data from 12 patients with left thalamic stroke who were tested on a series of recognition memory tasks (as well as other memory tests). High resolution structural MRI images were also obtained. The authors tested the hypothesis that the anterior thalamic nucleus / mammillo-thalamic tract (AN/MTT) are critical for recollection while the mediodorsal thalamic nucleus (MD) is critical for familiarity and possibly recollection too. It should be noted that the additional neuropsychological and neuroimaging data have already been reported (Danet, L., Barbeau, E. J., Eustache, P., Planton, M., Raposo, N., Sibon, I.,.… and Pariente, J. (2015). Thalamic amnesia after infarct The role of the mammillothalamic tract and mediodorsal nucleus. Neurology, 85(24), 2107-2115). This study concluded that lesions cause a severe amnesia in the case of MTT and to a lesser extent in the case of MD and that their study supports an updated view of the role of the MTT and MD in memory (Aggleton et al., 2011) suggesting that a disconnection of memory networks due to lesions to either a white matter tract (MTT) or to a specific nucleus (MD) may produce thalamic amnesia. Therefore, in my review I consider what this paper adds over and above the findings already reported from this cohort.

This issue of the neural substrates of recollection and familiarity has received a lot of interest over the past 20 years – without any clear consensus emerging. Potential limitations to progress are (4) whether psychological constructs such as recollection and familiarity can realistically be mapped on to neural circuits, (5) whether the tools used to quantify recollection and familiarity are adequate, and (2) whether the characterisation of recollection as a threshold process and familiarity as a continuous process is really valid. The authors sidestep these issues by attempting a straightforward test of the Aggleton et al., (2011) model, using three different procedures for estimating recollection and familiarity – all of which have been used extensively before.

The authors findings are partially inconsistent with the Aggleton et al., (2011) view. Damage to the MD and MTT produced an impairment in both recollection and familiarity (consistent with Aggleton et al., 2011). However, damage to the MD without additional damage to the MTT resulted in a mild impairment of recollection but not familiarity. To have shown the opposite result – a clear cut deficit in familiarity but not recollection following damage to MD alone would have been compelling evidence in favour of dual-process theory. However, given the findings already reported on these patients (who were impaired on verbal memory tasks; Danet et al., 2015), this was highly unlikely to be the case.

While these results will be of interest to researchers of recognition memory the discussion of the findings is all predicated on the fact that the tests used are reliable indices of recollection and familiarity. Recent research has strongly called this into question (e.g. Didi-Barnea C, Peremen Z, Goshen-Gottstein Y (2016) The unitary zROC slope in amnesics does not reflect the absence of recollection: critical simulations in healthy participants of the zROC slope. Neuropsychologia 90:94-109; Wixted JT, Mickes L, Squire LR (2010) Measuring recollection and familiarity in the medial temporal lobe. Hippocampus 20:1195-1205; Moran R, Goshen-Gottstein Y (2015) Old processes, new perspectives: Familiarity is correlated with (not independent of) recollection and is more (not equally) variable for targets than for lures. Cognitive Psychol 79:40-67). The key issue here is that the authors might fail to support Aggleton et al. (2011) because the processes of recollection and familiarity do not map on to the regions of thalamus in the way that Aggleton et al. propose or recollection and familiarity cannot be measured / dissociated using the methods the authors employ. These issues should be addressed in a revised version of the paper.

---

## [Author Response]

Reviewer #1:The question at the heart of this study is one that needs resolving. The neuropsychological approach is logical but inevitably poses interpretation problems based on precise lesion reconstructions. The authors have sensibly just looked at left thalamic strokes in order to examine the effects of MD and/or MTT damage. While the MTT results are a consistent loss of recollection with variable loss of familiarity (see Figure 3) the cases with MD but not MTT damage are notable for their sparing of familiarity – which seemingly contradicts predictions by Aggleton.The authors have gone to much length to detail the size and extent of the lesions (e.g. Supplementary file 2) as well as the figures in the main text. (It would be helpful to better flag the rather confusing case P1 re damage to MTT.)

P1 is indeed somewhat ambiguous since the two methods we used to assess the integrity of the MTT did not give congruent results. We chose to include this patient in the damaged MTT group because one of the methods indicated that the MTT was damaged in this patient. To be clearer about this specific case, and as suggested, we now have flagged P1 in Figure 2 and Figure 5 Note that this patient did not differ on the recognition memory tests from the rest of the patients included in the damaged MTT group.

The nonhuman primate work that provided the initial notion that MD has an important role in recognition (assumed to be familiarity) is barely considered or even referenced (see work of Gaffan, Parker, Aggleton, Mishkin, Squire) – though a review by Mitchell is considered. Why does this matter? Well those studies examined extensive cell loss in MDmc while the strokes in the present cases are more distributed between MDpc, MDmc and intralaminar n. (This pattern is quite typical of these kinds of strokes).

Thank you for this reminder. We now refer to this literature both in the Introduction (third paragraph) and the Discussion (fourth and twelfth paragraphs).

It would have been most helpful if coronal sections had been provided, so as to see how the strokes undercut MD and extend into MDpc. (I am assuming this is the pattern based on previous studies, but it’s incredibly important to know).

Indeed, we completely agree, and have added this data. Please see Figure 1—figure supplement 1.

For similar reasons, it is important (but incredibly difficult) to assess the amount of tract damage within the thalamus. These comments are not critical of the study but highlight why it is so hard to be definitive.

Again, we completely agree with this comment. Given current neuroimaging resolution in the human, this corresponds to a difficulty inherent to this type of studies. We now discuss this as a limitation of the study (last paragraph of the Discussion, copied below in our sixth response).

The authors mention that the thalamic group were impaired on verbal memory – which rather sets up the unsurprising result that as a group they were poor on recollective recognition – how does this relate to the two subgroups?

We previously had compared verbal memory in the damaged and intact MTT subgroups, showing more severe impairment in the damaged MTT subgroup (Danet et al., 2015). We now have added a sentence about this at the end of the Results section (“Subgroups comparisons” section).

Overall this is a well conducted study – my only concern is the difficulty of teasing apart the impact of the various nuclear/tract effects – which is a pervasive issue. The results do seemingly contradict some of the predictions made by Aggleton and Brown in 1999 (rather a straw man I am afraid) but they seem to align much closer to Aggleton et al. 2011 – with the 'intact' familiarity in the MTT intact group providing a new perspective, linked to the idea that familiarity reflects a much wider, diverse set of processes.

Thank you. As mentioned in our fourth response above, we now have more clearly acknowledged how difficult it is to assess nuclear and tracts damaged in such a small structure as the thalamus, as well as other possible caveats (last paragraph of the Discussion, copied below).

“It should be noted that more work needs to be done to better understand the involvement of the thalamus, and particularly of the MD, in memory. […] As noted earlier, future studies in patients could also favour tasks focusing specifically on familiarity processes (Migo *et al.,* 2009, Besson *et al.,* 2012).”

Reviewer #2:[…] However, I felt that the theoretical implications of the results should be discussed in more detail. This seems critical given there was a clear difference in the different patient groups and the fact that these results are not well described by the existing models. So, on the surface it would seem that the MD is critical for recollection, and that it is damage to the MTT that is critical for familiarity. I guess it just strikes me as odd that this is the case, and it seem worth speculating as to why this would be true.

Thank you for providing us with an opportunity to be clearer about this. We now explain our view in more detail in the Discussion.

In the second paragraph of the subsection “Recognition memory tasks”, it is concluded that "we therefore failed to find a dissociation between recollection and familiarity in any of the patients." This does not seem to be the case, however. As I understood the analysis the patients without involvement of the MTT did exhibit a selective impairment in recollection that left familiarity unaffected. So, this is a dissociation, in the traditional sense.

This is a misunderstanding. We were referring to dissociation at the single patient level, as an analysis at the group level may hide interesting findings. We rephrased this to be clearer in the second paragraph of the subsection “Recognition memory tasks”.

Reviewer #3:[…] While these results will be of interest to researchers of recognition memory the discussion of the findings is all predicated on the fact that the tests used are reliable indices of recollection and familiarity. Recent research has strongly called this into question (e.g. Didi-Barnea C, Peremen Z, Goshen-Gottstein Y (2016) The unitary zROC slope in amnesics does not reflect the absence of recollection: critical simulations in healthy participants of the zROC slope. Neuropsychologia 90:94-109; Wixted JT, Mickes L, Squire LR (2010) Measuring recollection and familiarity in the medial temporal lobe. Hippocampus 20:1195-1205; Moran R, Goshen-Gottstein Y (2015) Old processes, new perspectives: Familiarity is correlated with (not independent of) recollection and is more (not equally) variable for targets than for lures. Cognitive Psychol 79:40-67). The key issue here is that the authors might fail to support Aggleton et al. (2011) because the processes of recollection and familiarity do not map on to the regions of thalamus in the way that Aggleton et al. propose or recollection and familiarity cannot be measured / dissociated using the methods the authors employ. These issues should be addressed in a revised version of the paper.

We fully agree with this comment. We did our best to assess familiarity and recollection using well known, widely used, paradigms. Although as noted, they have been criticized. We do not think our results completely resolve the issues mentioned by reviewer 1 (the difficulty to precisely image the nuclei and tracts of the thalamus) and because of the difficulties, or limitations, to precisely assess familiarity and recollection you mentioned. With this regard, there may be paradigms focusing more precisely on familiarity that could prove useful in future work, such as those focusing on perceptual similarity (Migo et al., 2009) or on speed (by our own group, Besson et al., 2012; 2015; Barragan-Jason et al., 2014). Dedicated structural MRI sequences, rather than state of the art but standard ones, could also be developed in the near future, not to mentioned 7T imaging. Altogether, these advances could critically help with these issues in the future.

However, given that we used three different paradigms to assess recollection and familiarity (and as precise structural neuroimaging as we could) we think our results, together with the literature we report, are convergent and strong. Our findings therefore call for a revision of Aggleton et al., 2011 regarding familiarity.

To be clearer about all this, we added a paragraph (last paragraph of the Discussion, copied below).

“It should be noted that more work needs to be done to better understand the involvement of the thalamus, and particularly of the MD, in memory. […] As noted already, future studies in patients could also favour tasks focusing specifically on familiarity processes (Migo et al., 2009, Besson et al., 2012).”